

# Historical museum collections and contemporary population studies implicate roads and introduced predatory bullfrogs in the decline of western pond turtles

E. Griffin Nicholson[1], Stephanie Manzo[1], Zachary Devereux[1], Thomas Paul Morgan[1], Robert N. Fisher[2], Christopher Brown[2], Rosi Dagit[3], Peter A. Scott[4] and H. Bradley Shaffer[4,5]

[1] Institute of the Environment and Sustainability, University of California, Los Angeles, CA, USA
[2] U.S. Geological Survey, Western Ecological Research Center, San Diego, CA, USA
[3] Resource Conservation District of the Santa Monica Mountains, Topanga, CA, USA
[4] Department of Ecology and Evolutionary Biology, University of California, Los Angeles, CA, USA
[5] La Kretz Center for California Conservation Science, Institute of the Environment and Sustainability, University of California, Los Angeles, CA, USA

Corresponding author
E. Griffin Nicholson,
gnicholson@g.ucla.edu

## ABSTRACT

The western pond turtle (WPT), recently separated into two paripatrically distributed species (*Emys pallida* and *Emys marmorata*), is experiencing significant reductions in its range and population size. In addition to habitat loss, two potential causes of decline are female-biased road mortality and high juvenile mortality from non-native predatory bullfrogs (*Rana catesbeiana*). However, quantitative analyses of these threats have never been conducted for either species of WPT. We used a combination of historical museum samples and published and unpublished field studies shared with us through personal communications with WPT field researchers (B. Shaffer, P. Scott, R. Fisher, C. Brown, R. Dagit, L. Patterson, T. Engstrom, 2019, personal communications) to quantify the effect of roads and bullfrogs on WPT populations along the west coast of the United States. Both species of WPT shift toward increasingly male biased museum collections over the last century, a trend consistent with increasing, female-biased road mortality. Recent WPT population studies revealed that road density and proximity were significantly associated with increasingly male-biased sex ratios, further suggesting female-biased road mortality. The mean body size of museum collections of *E. marmorata*, but not *E. pallida*, has increased over the last 100 years, consistent with reduced recruitment and aging populations that could be driven by invasive predators. Contemporary WPT population sites that co-occur with bullfrogs had significantly greater average body sizes than population sites without bullfrogs, suggesting strong bullfrog predation on small WPT hatchlings and juveniles. Overall, our findings indicate that both species of WPT face demographic challenges which would have been difficult to document without the use of both historical data from natural history collections and contemporary demographic field data. Although correlational, our analyses suggest that

female-biased road mortality and predation on small turtles by non-native bullfrogs are occurring, and that conservation strategies reducing both may be important for WPT recovery.

## INTRODUCTION

Natural history museum collections are often the only source of historical information for declining and endangered species, and can be an important tool when documenting and analyzing species declines and their causes. Specimen data have primarily been used to determine changes in the distribution of individual species or taxonomic groups (*Shaffer, Fisher & Davidson, 1998*; *Pyke & Ehrlich, 2010*), often focusing on species of conservation concern (*Fisher & Shaffer, 1996*; *Grixti et al., 2009*; *Hamer, Lane & Mahony, 2010*; *Major & Parsons, 2010*; *Saarinen & Daniels, 2012*). Although less commonly emphasized, museum collections can also provide historical insights into demographic changes that are important proximate mechanisms of population trajectories and necessary for population viability analyses (PVAs, *McCarthy, Burgman & Ferson, 1995*; *Lacy, 2000*). This is particularly important for declining species being considered for formal protection, given that such conservation actions often rest on evidence of population trends. If museum collections are unbiased samples of species through time (a strong assumption in many cases), they can provide critical clues about population and demographic histories. For example, museum specimens have recently been utilized for multiple species as evidence of shifts in body size due to climate change (*Babin-Fenske, Anand & Alarie, 2008*; *Caruso et al., 2014*, *Weeks et al., 2020*).

We conducted a comprehensive analysis of museum specimens of western pond turtles (WPTs, Fig. 1), a pair of species which is currently petitioned for listing under the U.S. Endangered Species Act (*U.S. Fish and Wildlife Service, 2015*). Habitat loss and fragmentation have been identified as major factors associated with population declines in WPTs, particularly in Southern California (*Thomson, Wright & Shaffer, 2016*). In addition, perceived, but largely unstudied demographic changes have led to two further hypotheses for the decline of both WPT species.

The first hypothesis centers on observations that the sex ratio of WPT populations is often male-biased (*Spinks et al., 2003*; *Polo-Cavia et al., 2010*; although see *Germano & Riedle, 2015*), leading to the speculation that terrestrial roadkill mortality has preferentially removed nesting adult females from many populations. In the United States, a relatively recent review of the literature on turtle population sex ratios, published between 1928 and 2003 covering 36 different species, reported an average 22% increase in the intrapopulation proportion of males, and that this bias was most pronounced in aquatic species (*Gibbs & Steen, 2005*). Other studies have found similar increases in male bias in turtle populations (*Aresco, 2005*; *Vanek & Glowacki, 2019*). Although this body of work

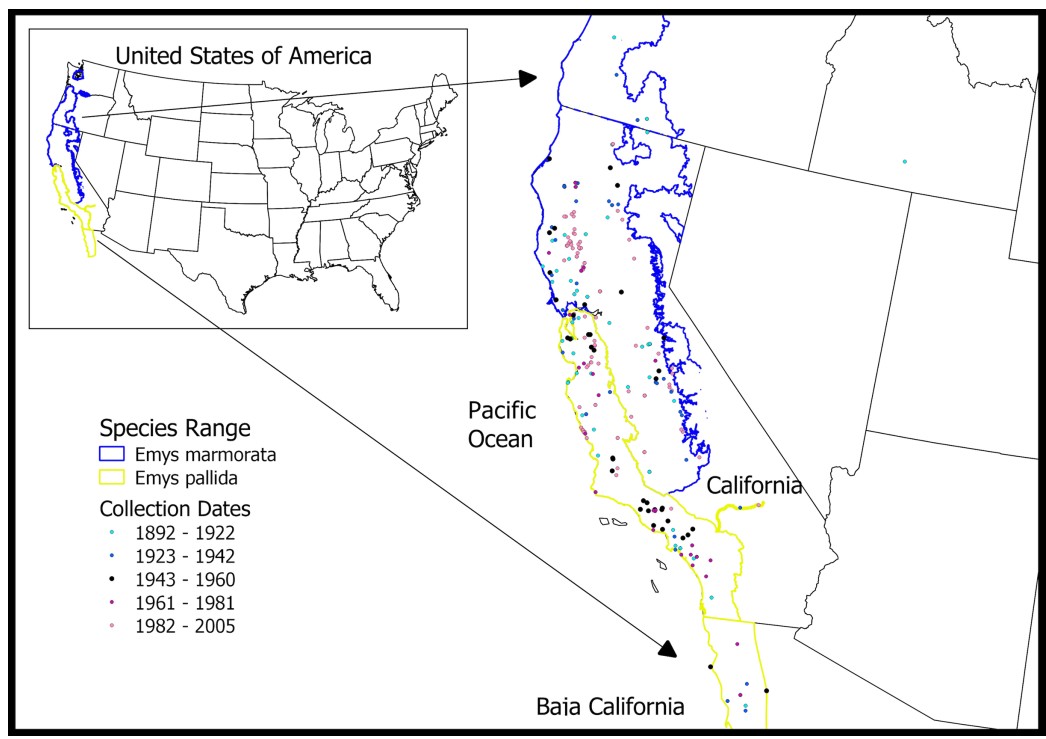

**Figure 1 WPT museum specimens included in this study.** Collection dates range from 1892 to 2005. Species ranges from *Gogol-Prokurat (2016)* (for California), USGS Gap Analysis Project, 2017 (for Oregon and Washington), and SDNHM HerpAtlas, 2014 (for Baja California, Mexico).

clearly indicates an increase in male bias across many aquatic turtle species, there is less certainty about the cause of these shifts. Roads and their associated vehicle traffic can have profound impacts on wildlife (*Spellerberg, 1998*), and studies of other species have suggested that vehicular mortality results in male biased turtle populations near roads (*Marchand & Litvaitis, 2004*; *Steen & Gibbs, 2004*; *Aresco, 2005*). Countering this, a study of painted turtles (*Chrysemys picta*) near Chicago, IL found a large male bias (75% male), but no evidence of road density as a causal factor (*Vanek & Glowacki, 2019*), and a population model predicted that populations of small-bodied pond turtles, like the WPT, should not be threatened by road mortality anywhere in the United States (*Gibbs & Shriver, 2002*). In Ontario, Canada, one study of painted turtles found no higher frequency of males at sites closer to major roads than more remote sites further from roads (*Dorland, Rytwinski & Fahrig, 2014*). Another study in the Ontario area used long-term data on turtles admitted to the Kawartha Turtle Trauma Centre and found that Midland Painted Turtles (*Chrysemys picta marginata*), Snapping Turtles (*Chelydra serpentina*), and Blanding's Turtles (*Emydoidea blandingii*) did not have significantly biased sex ratios among admissions for road injuries. However, Northern Map Turtle (*Graptemys geographica*) road injury admissions were significantly female-biased (*Carstairs, Dupuis-Desormeaux & Davy, 2018*), indicating that the impact of road mortality on sex ratios may differ between sympatric species on the same landscape. Nesting WPTs may be

especially vulnerable to vehicular mortality because they can travel hundreds of meters from their aquatic habitats to find nesting sites (*Storer, 1930*; *Holland, 1994*). However, evidence for road-proximity caused shifts in sex ratio in the WPT is limited to two reports (*Holland, 1994*; *Madden-Smith et al., 2005*), both of which suggest that vehicular traffic may be leading to differential female mortality. A more comprehensive analysis is needed to specifically evaluate the hypothesis that roads are the cause of an increase in male bias in WPTs.

A second hypothesis for the decline of WPTs comes from ecological surveys suggesting reduced juvenile recruitment due to predation by non-native bullfrogs (*Lovich et al., 2017*; *Smith, 2018*). American bullfrogs (*Rana catesbeiana*) are a pervasive invasive species outside of their native eastern U.S. range and are opportunistic predators on many aquatic vertebrates. An examination of bullfrog stomach contents from southern Vancouver Island, British Columbia, Canada found that 6% of total prey were western painted turtle (*C. picta bellii*) hatchlings (*Jancowski & Orchard, 2013*), demonstrating that these large amphibians regularly consume small turtles in their non-native range. However, in their native range, bullfrogs are sympatric with most North American turtles and turtle hatchlings comprise a consistent, but apparently minor component of their diet (*Bury & Whelan, 1984*). At least one study found no evidence of significantly higher bullfrog densities or average individual sizes in introduced populations (*Govindarajulu, Price & Anholt, 2006*), suggesting that the cause of turtle declines by bullfrogs may be due to the behavior of turtles in historically bullfrog-free regions. Individuals that did not co-evolve with this novel predator may lack appropriate anti-predator responses, particularly in naïve baby turtles that are most susceptible to predation. Invasive bullfrogs are now found across much of the range of the WPT (Fig. S1; *Holland, 1994*), and documented predation has led several authors to hypothesize that they are responsible for WPT declines and extirpations (*Lovich & Meyer, 2002*; *Hallock, McMillan & Wiles, 2017*; *Lovich et al., 2017*; Fig. 2). Hatchlings and small juveniles less than 3 years old are the most vulnerable to predation by bullfrogs (*Hallock, McMillan & Wiles, 2017*). Compelling, but still anecdotal, information on the effect of bullfrogs was highlighted in a study in northwestern California, USA, where four out of six lentic sites near the Trinity River had abundant bullfrog populations and were biased towards large, old WPTs, while two sites lacking bullfrogs did not exhibit this trend (*Sloan, 2012*). Although much of the research on the impacts of bullfrogs on WPT populations has been anecdotal, the number of studies (*Lovich & Meyer, 2002*; *Sloan, 2012*; *Hallock, McMillan & Wiles, 2017*; *Lovich et al., 2017*) finding a correlation between bullfrog presence and a decrease in juvenile turtles suggests that bullfrogs could be a serious threat to the persistence and viability of WPT populations.

Our goal in this study was to quantitatively assess trends in shifting sex ratios and population demography in WPTs, and the potential roles of road mortality and predation by non-native American bullfrogs as causal agents (*Lovich & Meyer, 2002*; *Hallock, McMillan & Wiles, 2017*; *Smith, 2018*). We analyzed the majority of available museum specimens, combined these historic records with several published and unpublished data sets of contemporary populations shared with us through personal communications with WPT field researchers (B. Shaffer, P. Scott, R. Fisher, C. Brown, R. Dagit, L. Patterson,

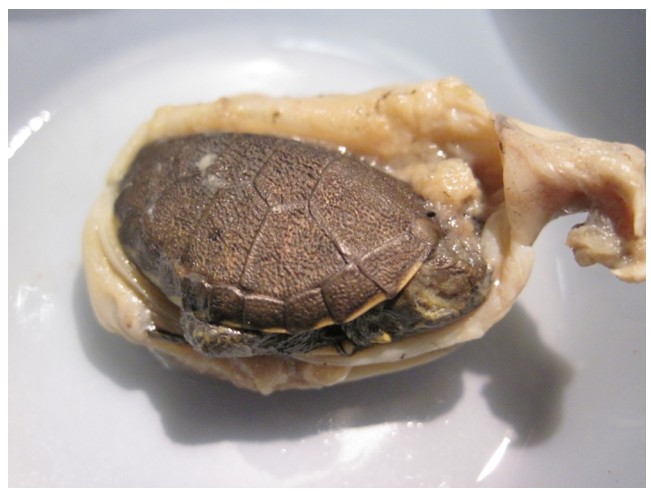

**Figure 2 Hatchling western pond turtle found in the stomach of a bullfrog in the San Luis Rey River, San Diego Co. USA.** Source: U.S. Geological Survey, Western Ecological Research Center, San Diego, CA, USA.                                                    

T. Engstrom, 2019, personal communications), and quantified historical trends in the WPT from the earliest to the most current information available. We used these observational data sets to quantify trends in sex ratio and juvenile recruitment over time, and to ask whether road density and the presence of bullfrogs were plausible causal factors responsible for those trends. Although these tests are correlational, our analyses strongly suggest that road density and bullfrog occupancy may be important causal agents of WPT population declines.

## MATERIALS AND METHODS

### Taxonomy

The WPT complex comprises the only extant native freshwater turtles in California (*Thomson, Wright & Shaffer, 2016*; *Turtle Taxonomy Working Group, 2017*). Until recently, the WPT was considered a single polytypic species ranging from Baja California, Mexico to Washington State, USA that has variously been classified in the genus *Emys*, *Actinemys*, or *Clemmys* (*Fritz, Schmidt & Ernst, 2011*; *Turtle Taxonomy Working Group, 2017*). The generic assignment of the WPTs is still controversial, with vocal proponents placing the two species in a more inclusive *Emys* or a more narrowly defined *Actinemys* (*Fritz, Schmidt & Ernst, 2011*; *Spinks et al., 2016*; *Turtle Taxonomy Working Group, 2017*); we follow *Spinks, Thomson & Shaffer (2014)* and *Spinks et al. (2016)* in recognizing a more inclusive *Emys* as the generic allocation. In addition, recent multi-locus molecular genetic analyses indicate that the WPT consists of two distinct, geographically non-overlapping species: *Emys (Actinemys) pallida* and *E. (A.) marmorata* (*Spinks, Thomson & Shaffer, 2014*). *Emys pallida* occupies the southern and coastal portion of the range from Baja California to roughly San Francisco Bay, while *E. marmorata* occupies the inland San Joaquin Valley/Sierra Nevada foothills of Central California north to Washington State (*Thomson, Wright & Shaffer, 2016*; Fig. 1).
Throughout this article, when we refer simply to WPTs without indicating species, we are considering both of these ecologically similar species. However, for many of our analyses, we differentiate the two species so that each can be evaluated across its range.

## Museum samples

We surveyed WPT specimens from three museums to assess changes in sex ratios and carapace (shell) length over time. We evaluated a total of 463 individual WPT specimens collected from 44 counties between 1892 and 2005 (the Natural History Museum of Los Angeles County, Los Angeles, CA ($n = 55$), California Academy of Sciences, San Francisco, CA ($n = 151$), and Museum of Vertebrate Zoology, Berkeley, CA ($n = 257$); Fig. 1; Dataset S1). This represents 82% (463/566) of the available alcohol/ethanol preserved WPT specimens on VertNet (http://portal.vertnet.org/search; search terms Emys/Actinemys/Clemmys pallida/marmorata, alcohol/ethanol, basisOfRecord = PreservedSpecimen (accessed on 2019-08-11)). We measured midline carapace length for each specimen (Fig. S2; *Iverson, 2018*) and recorded the sex of each specimen, based on tail length and plastral curvature. Other key information including collection date, county, state and GPS coordinates were extracted from online museum databases. Using distribution information from the most recent molecular analysis (*Spinks, Thomson & Shaffer, 2014*), we classified each specimen as either *E. pallida* ($n = 147$) or *E. marmorata* ($n = 316$) based on its county of origin (Fig. 1; Table S1).

## Contemporary samples

Demographic information for current samples primarily came from unpublished data contributed by WPT field researchers (Fig. 3). For *E. pallida*, data sets came from trapping data provided by the USGS, San Diego office (R. Fisher & C. Brown, 2019, personal communications), the Resource Conservation District of the Santa Monica Mountains, Los Angeles County, CA (R. Dagit, 2019, personal communication), Lake Elizabeth, Los Angeles County, CA (B. Shaffer, 2019, personal communication) and a trapping survey in the Jack and Laura Dangermond Preserve, Santa Barbara County (B. Shaffer & P. Scott, 2019, personal communications). Field tagging and tracking of WPTs in the Santa Monica Mountains was approved by the California Department of Fish and Wildlife (CDFW SC-604). USGS sampling and field tagging of WPTs in Southern California was also approved by the CDFW (CDFW SCP-838). The Nature Conservancy gave permission to sample on the Dangermond Preserve which was also approved by the CDFW (CDFW SC-2480). For *E. marmorata*, data sets included two sites near Chico, Butte County, CA (T. Engstrom, 2019, personal communication), Sacramento County, CA (L. Patterson, 2019, personal communication), and the University of California Davis Arboretum, Yolo County, CA (*Lambert et al., 2019*). We extracted the midline carapace length (a standard index of body size), sex, date of capture, location and population size from each study.

## Sex ratio

All turtles with straight-line carapace length less than 110 mm were considered immature and removed from the sex ratio analysis for both the museum and current samples.

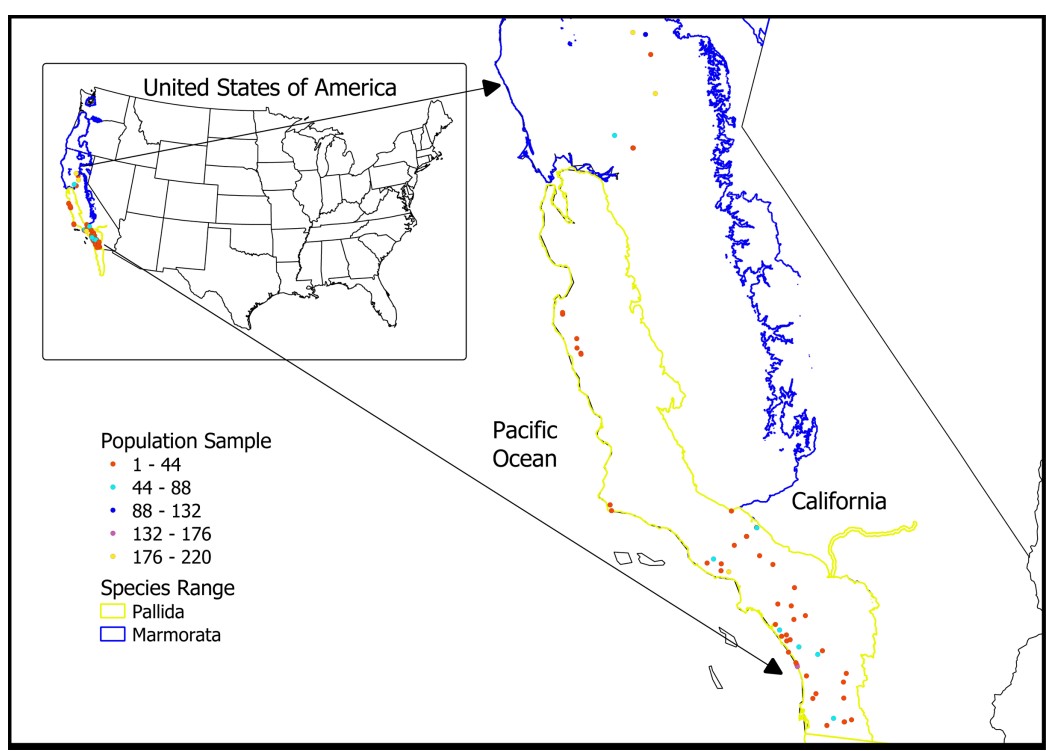

**Figure 3 WPT contemporary population samples collected between 2002 and 2019.** Sources: R. Fisher & C. Brown, 2019, personal communications; R. Dagit, 2019, personal communication; B. Shaffer, 2019, personal communication; B. Shaffer & P. Scott, 2019, personal communications; T. Engstrom, 2019, personal communication; L. Patterson, 2019, personal communication; *Lambert et al., 2019*.

This cutoff is the typical size at which WPTs reach sexual maturity and secondary sexual characteristics become apparent (*Holland, 1991*), although for southern California *E. pallida* it may be a few mm smaller (*Madden-Smith et al., 2005*). To quantify changes in sex ratio, we used logistic regression (RStudio Version 1.2.1335; *R Studio Team, 2018*; *R Core Team, 2019*) to plot the probability of sex through time for *E. pallida* and *E. marmorata* separately. For each species, three analyses were conducted: a historical regression based on the museum samples only, a regression based on only contemporary samples, and a combined regression based on both the museum and current samples. Within the contemporary data sets, which often recorded the same turtle multiple times within and between years, we counted each individual only once per year. We reasoned that across years recaptured individuals should be included in the sex ratio for each year because that individual survived, and therefore contributes to the sex ratio each year.

## Carapace length

Given that the contemporary data sets had WPTs which were recaptured multiple times, we filtered the data so that each individual was represented only by its most recent capture data. We reasoned that to make the museum and recent data as comparable as possible, we should only use the last capture as an estimate of body size since the museum

specimens could be considered the last capture of each preserved turtle. After filtering the contemporary data sets, we combined these data with our museum specimen data.

To quantify changes in carapace length, we used linear regression in R (RStudio Version 1.2.1335; *R Studio Team, 2018*; *R Core Team, 2019*) to plot carapace length through time for *E. pallida* and *E. marmorata* separately. The linear regression was considered significant at a *p*-value < 0.05 and the adjusted $R^2$ a measure of the variance explained by that relationship.

## The effects of roads on sex ratios

To analyze the effects of roads on sex ratios, we conducted two independent analyses on the same set of WPT population sites. Based on contemporary studies from the last 20 years, we calculated the sex ratios of WPT population studies that included data on at least 10 unique turtles in each field survey or study (*Gibbs & Steen, 2005*). Some sites were the subject of long-term monitoring studies, and we used the most recent 7 years of data (corresponding to the average number of years to sexual maturity, *Belli, 2015*) to estimate the sex ratio for these sites. We used the latitude and longitude of each site to create data points in ArcGIS, mapped them to a 2018 USGS topographic map (USGS Topo), and created an aquatic habitat polygon for each site. For a pond or lake, the polygon was simply the extent of the body of water. For a river, creek, or stream, the polygon included the river channel spanning the two most distant turtle captures.

We first analyzed the relationship between population sex ratio and distance to the nearest road. We calculated and plotted the centroid of each aquatic habitat polygon in ArcGIS (*ESRI, 2018*, Version 10.6.1, Calculate Geometry option); if a centroid was on land, we manually moved it to the nearest edge of the aquatic habitat polygon, and manually measured the shortest straight-line distance from the centroid to the closest road. Centroids that were further than 1 km from the nearest road were not included in the analysis because the maximum distance that nesting female WPTs have been recorded from water is approximately 400 m (*Storer, 1930*; *Holland, 1994*); although this number is approximate, we reasoned that sites very distant from roads would suffer little or no road mortality. We used linear regression and a "sliding scale *t*-test", described below, to describe the relationship between population sex ratio and distance to the closest road.

We also analyzed the relationship between population sex ratio and the density of surrounding roads. We used ArcGIS to create a 400 m buffer surrounding each aquatic habitat polygon. We then created polylines of the roads within the buffer region of each population site, buffered the polyline feature class 6 m, and summed the total buffered area. This gave a total road surface area, assuming that each road is two lanes (one in each direction) and is slightly over the 2.7 m minimum allowable width (*U.S. Department of Transportation Federal Highway Administration, 2013*). For the three population sites which had multi-lane highways within their 400 m buffer, we considered the highways as distinct from the road polyline and buffered each based on the actual number of lanes (3 m/lane). After all polyline buffering was complete, the fraction of the 400 m buffered polygon area covered by roads was calculated as the area of the buffered polyline

(including highway area if present) divided by the area of the buffered polygon after subtracting the area of the contained water body. We used linear regression to describe the relationship between total road area and population sex ratio.

### The effects of bullfrogs on average WPT size/age

To quantify the effects of introduced predatory bullfrogs on hatchling and juvenile survival, we compared size distributions of WPT populations that do or do not have bullfrogs present. Given the anecdotal information supporting the negative impact of bullfrogs on WPT populations (e.g., Fig. 2), we focused on bullfrogs rather than other potential predators (largemouth bass, wading birds) because they are visible, sedentary predators and their presence is easily determined. However, we note that bullfrog presence is often highly correlated with other introduced predators including sunfish, bass and crayfish (*Fisher & Shaffer, 1996*; *Madden-Smith et al., 2005*; *Riley et al., 2005*; *Miller et al., 2012*), and distinguishing the specific effect of bullfrogs from other invasive species is impossible with observational data. For each contemporary data set that had at least 10 unique individual turtles, we calculated the average carapace length of the population for the most recent 7 years, used a combination of published reports and personal communication with field biologists to confirm the presence or absence of bullfrogs at each site and conducted *t*-tests (two tailed) on carapace lengths of sites with/without bullfrogs. One site in San Diego County that has undergone bullfrog eradication measures was treated as two sites, one for pre-eradication and one for post-eradication. For population sites where bullfrog observations were not available, we queried iNaturalist and recorded bullfrogs as present if a sighting had been recorded within 1 km of the WPT population site.

## RESULTS

### Sex ratio changes through time

#### Emys pallida

Based on the combined historical museum data and contemporary data sets (Fig. S3), the probability of observing female specimens shifted from ~0.5 in 1900 to approximately 0.4 in contemporary samples (Fig. 4, Hosmer and Lemeshow goodness of fit (GOF) test, $p = 0.0005$). The historical museum data alone show a similar, but non-significant decrease in the probability of female specimens over time, while the contemporary data set indicates a slight but significant increase in the probability of female captures (Fig. 4; ~0.4 to ~0.5).

#### Emys marmorata

The trends in sex ratio probabilities over time in *E. marmorata* were similar to those in *E. pallida*. From the combined historical museum data and current data sets (Fig. 4), the overall probability of observing females in *E. marmorata* collections shifted from ~0.5 in 1900 to a more male-biased sex ratio of ~0.4 in contemporary populations (Fig. 4, Hosmer and Lemeshow goodness of fit (GOF) test, $p = 0.0003$). The historical museum data and contemporary population surveys show similar trends over time (Fig. 4).

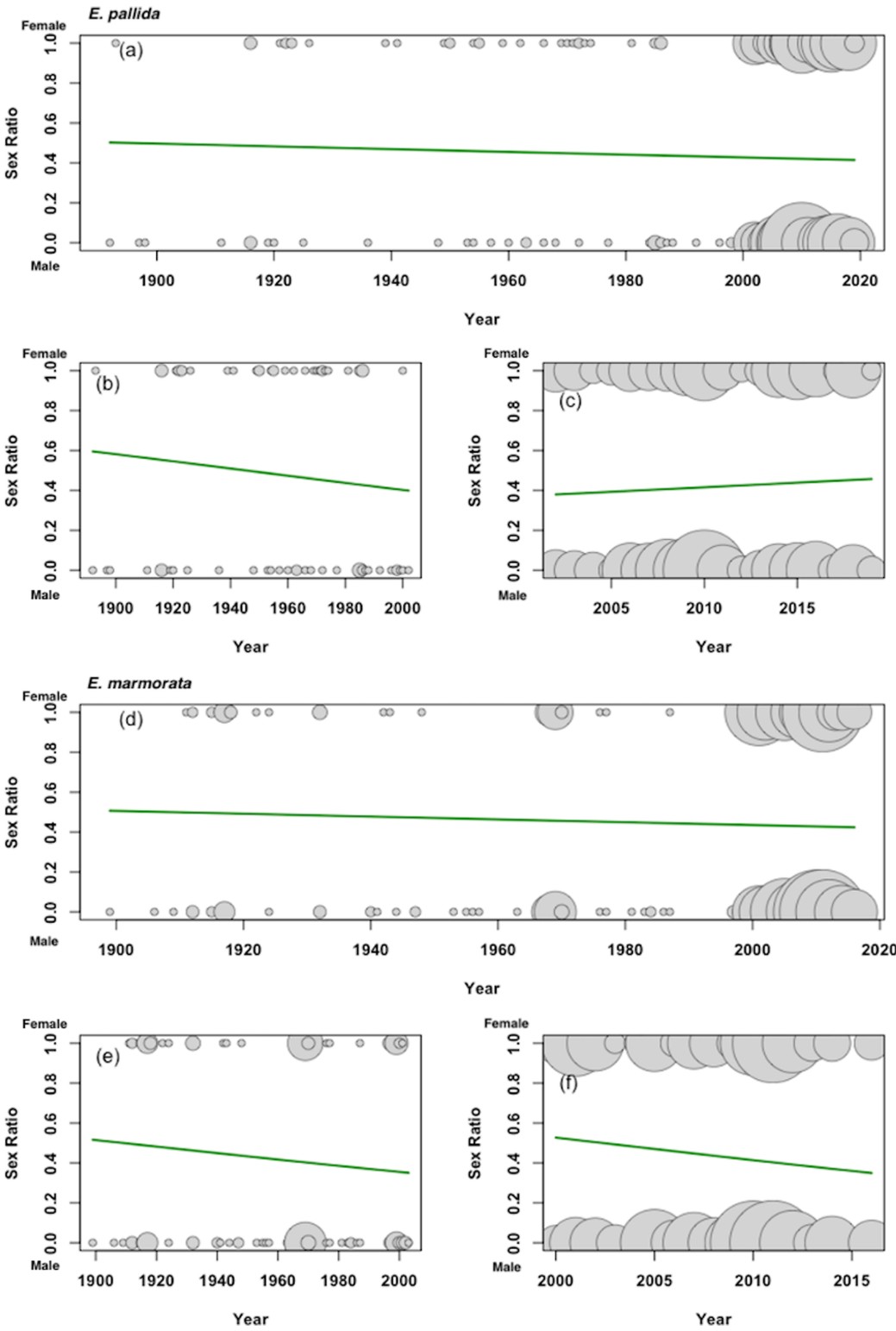

**Figure 4** **Logistic regressions of sex ratio for *E. pallida* and *E. marmorata*.** Female WPTs were scored as 1 whiles males were scored as 0. The sample sizes for each year are represented by gray circles which are proportional to the sample size for the year. (A) Combined regression based on both the *E. pallida* museum and current samples (Hosmer and Lemeshow goodness of fit (GOF) test: *X*-squared = 27.871, df = 8, *p*-value = 0.0004994). (B) Regression based on the *E. pallida* museum samples (Hosmer and

**Figure 4** (continued)
Lemeshow goodness of fit (GOF) test: $X$-squared = 11.625, df = 8, $p$-value = 0.1687). (C) Regression based on the current *E. pallida* samples (Hosmer and Lemeshow goodness of fit (GOF) test: $X$-squared = 19.911, df = 8, $p$-value = 0.01068). (D) Combined regression based on both the *E. marmorata* museum and current samples (Hosmer and Lemeshow goodness of fit (GOF) test: $X$-squared = 29.457, df = 8, $p$-value = 0.0002636). (E) Regression based on the *E. marmorata* museum samples (Hosmer and Lemeshow goodness of fit (GOF) test: $X$-squared = 2.6945, df = 8, $p$-value = 0.952). (F) Regression based on the current *E. marmorata* samples (Hosmer and Lemeshow goodness of fit (GOF) test: $X$-squared = 32.3, df = 8, $p$-value = 8.299E−05).

## Carapace length

### Emys pallida

Mean carapace length of *E. pallida* remained relatively constant from 1892 to 2019 (significance test of linear regression, $p$ = 0.43) (Fig. 5). Museum samples (1892–2005) considered alone showed a slight, but non-significant (significance test of linear regression, $p$ = 0.17) increase in carapace length over time (Fig. S5). Thus, there is no evidence that body size has been increasing as would be the case if there has been increasing juvenile mortality over time.

### Emys marmorata

Unlike *E. pallida*, mean carapace length increased significantly in *E. marmorata* from 1894 to 2016 (significance test of linear regression, $p$ = <2.2E−16) (Fig. 5). This trend holds with and without contemporary populations (museum specimens from 1892 to 2005, significance test of linear regression, $p$ = 0.00056, Fig. S6) suggesting that the trend is not solely a function of the recent population data. Together, the museum data and current data sets indicate that overall body size has been increasing in *E. marmorata* over the last century.

## The effects of roads on sex ratio

We pooled both species of WPT for our more detailed examination of road mortality as a cause of male-biased sex ratios. We did so for two reasons. First, no studies have indicated that the two species differ in the distance females travel from a body of water to nest, although this is admittedly a very understudied aspect of WPT biology (*Brehme, Hathaway & Fisher, 2018*). Second, the number of population samples available for each species was relatively small, and pooling both taxa allowed us greater statistical power to detect trends.

When pooled across all sites and distances, the sex ratio of WPT populations showed a negative, but non-significant relationship to distance to the nearest road (significance test of linear regression, $p$ = 0.13, Fig. S7). However, visual inspection suggested that there was a shift toward less female-biased sex ratios when the closest road was approximately 250 m or more distant from the centroid of the body of water (Fig. S7). To quantitatively explore the existence and location of a sex-ratio shift in the data, we conducted a "sliding-scale $t$-test" for the 19 population samples in our analysis. We first tested for a significant difference in the sex ratio of the two populations closest to roads

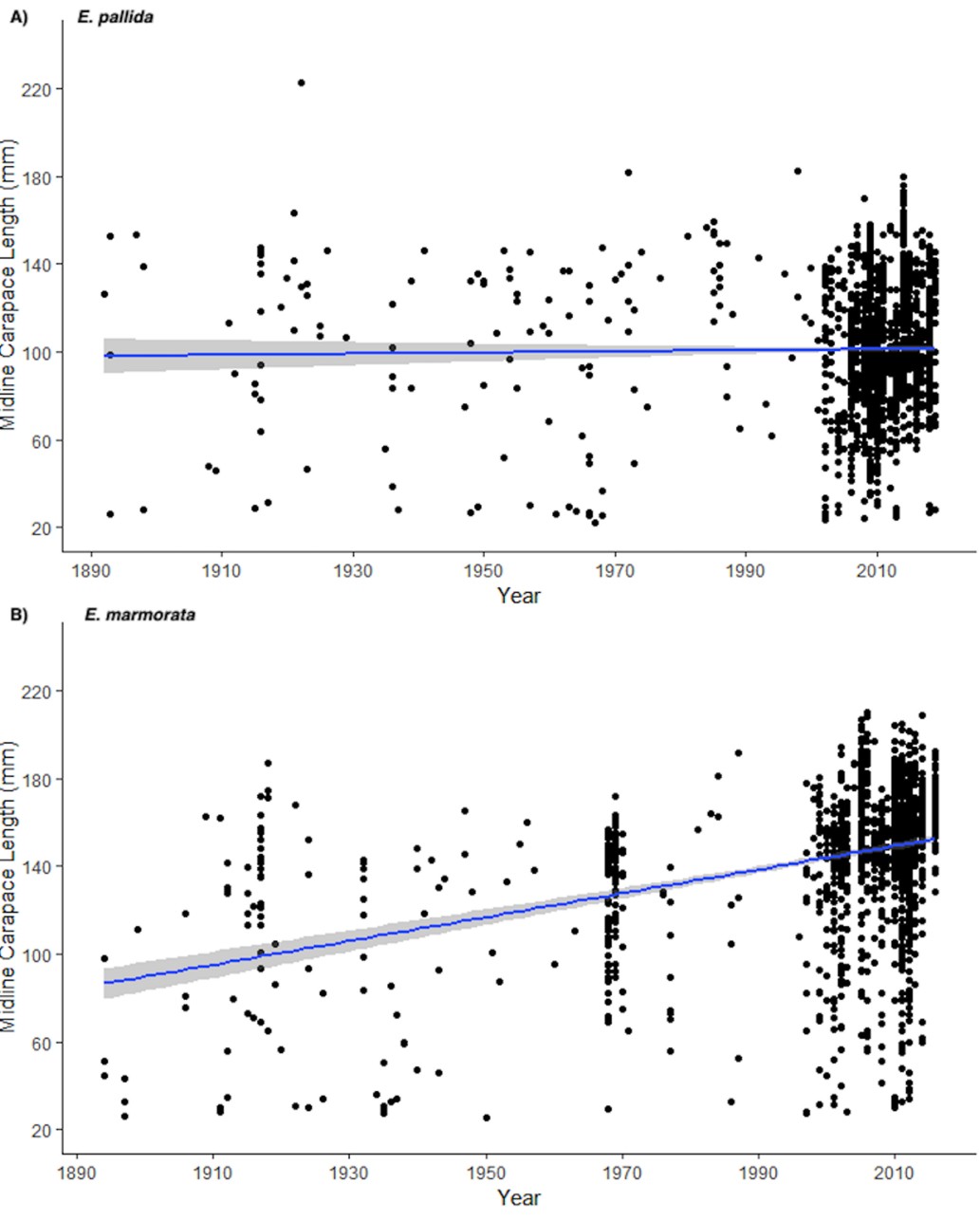

**Figure 5 Plots of midline carapace lengths vs. time for (A) *E. pallida* from 1892 to 2019 (*n* = 2095, Adjusted *R*-Squared = −0.0001874, *p* = 0.43) and for (B) *E. marmorata* from 1894 to 2016 (*n* = 1697, Adjusted *R*-Squared = 0.13, *p* = < 2.2E−16).** The blue trend lines and grey shading show the linear regressions and 95% confidence intervals for the slopes of the lines.

(we refer to this as the road-proximate set of sites) compared to the 17 more distant sites. We then added the third closest site to the road-proximate set and performed a new *t*-test; we iteratively continued until the sex ratio of the road-proximate set differed from the remaining more distant populations. We found that the shift in significance occurred at 219 m from the centroid of the body of water (Table S2). Sites where the nearest road was

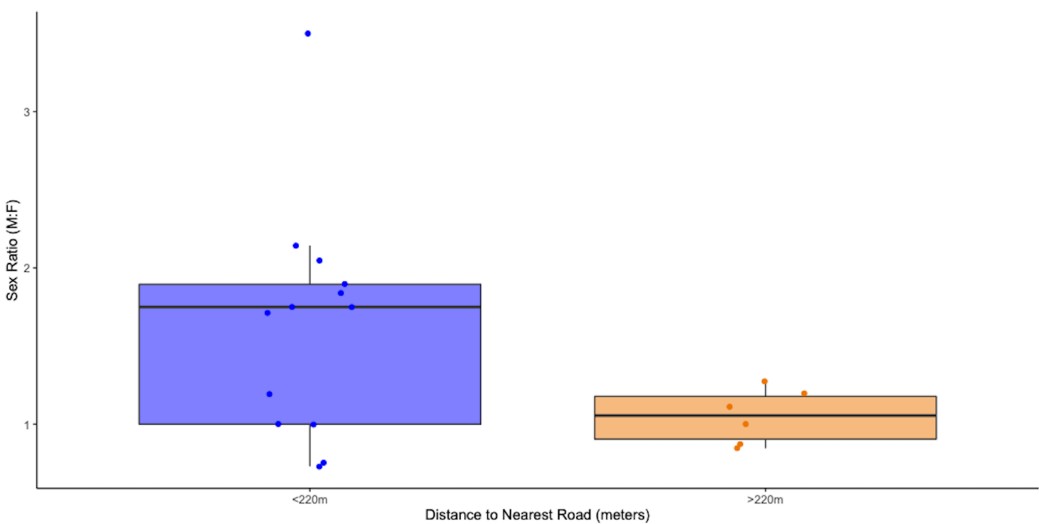

**Figure 6 Plotted comparison of the sex ratios (Male: Female) of WPT population sites with the nearest road under 220 m away with sites with the nearest road over 220 m away ($n$ = 13 for under 220 m, $n$ = 6 for over 220 m, $p$ = 0.016909056).**

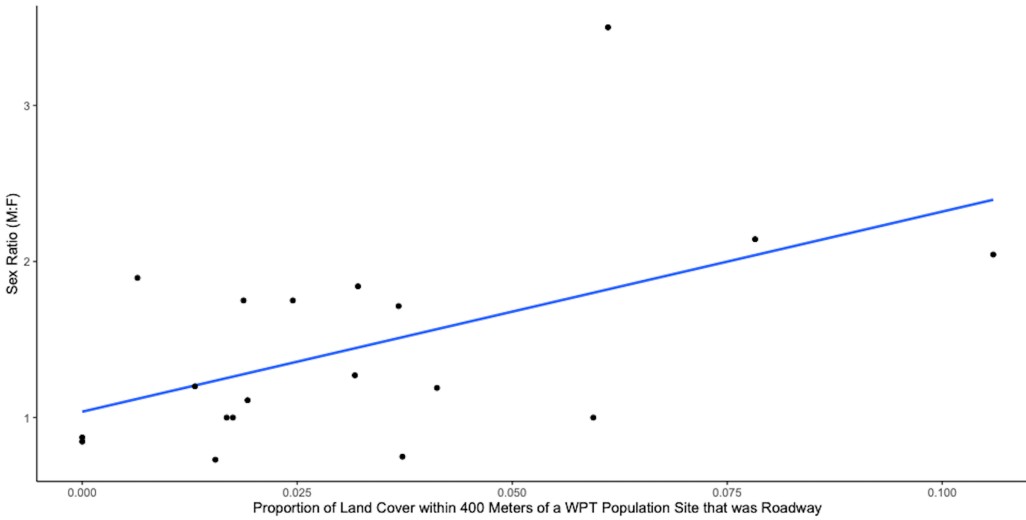

**Figure 7 Plot of sex ratio (Male: Female) vs. proportion of land cover within 400 m of a WPT population site that was roadway ($n$ = 19, $R$-squared = 0.2698, $p$ = 0.022663).**

closer than 219 m were significantly more male biased (average sex ratio ~1.64) than those more distant from the nearest road, which were tightly clustered around an even sex ratio (Fig. 6, two tailed $t$-test, $p$ = 0.017).

Road density within a biologically reasonable buffer may be a more important indicator of potential vehicular mortality than simply distance to the nearest road. The sex ratio of WPT populations became increasingly male biased as the proportion of land covered by roads within 400 m of a water body increased, explaining 27% of the variance in adult sex ratio (Fig. 7, significance test of linear regression, $p$ = 0.023).

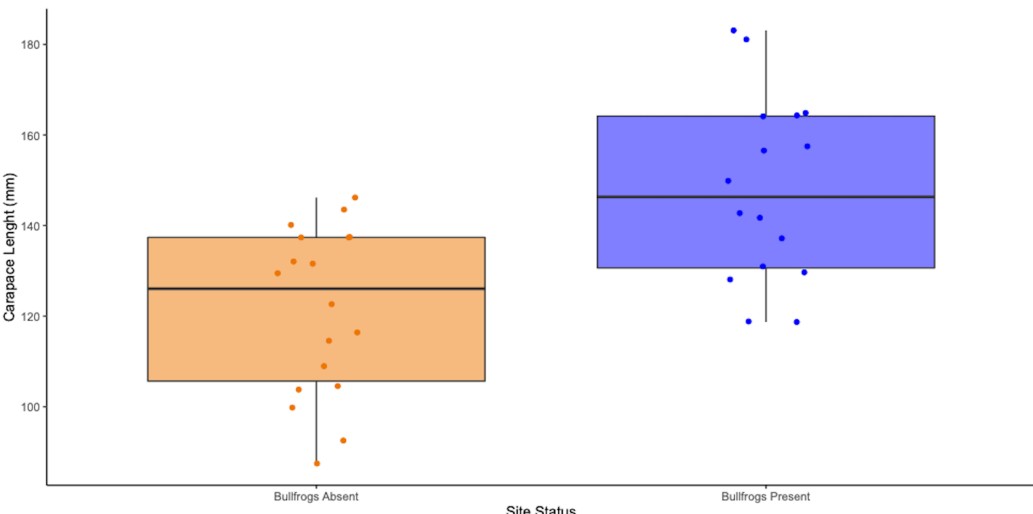

**Figure 8** Average carapace lengths (mm) of WPT population sites with bullfrogs absent (*n* = 18) and present (*n* = 16) pooled across *E. marmorata* and *E. pallida.* The difference is significant (two-tailed *t*-test, *p* = 0.00039337).

## The effects of bullfrogs on body size distribution

We also pooled the bullfrog data for both species of WPT. We found a significant difference (two-tailed *t*-test, df = 30.525, *p* = 0.00040) between the average carapace length of population sites with bullfrogs co-occurring and population sites without bullfrogs (Fig. 8). Population sites with bullfrogs present had an average carapace length of 148 mm, while those without this introduced predator had an average carapace length of 122 mm (Fig. 8). However, there is a potential confounding factor caused by pooling the two WPT species because *E. marmorata* individuals are typically larger than *E. pallida* (Fig. 5). When conducted for each species separately, we found the same general trend in both species. Sites with bullfrogs had higher mean carapace length and fewer smaller individuals, and this trend was significant for both *E. pallida* (two-tailed *t*-test, *p* = 0.0444) and *E. marmorata* (two-tailed *t*-test, *p* < 0.004, Figs. S8 and S9).

## DISCUSSION

As a group, turtles are arguably the most threatened "major" vertebrate clade, with 52–61% of modern turtle species threatened or already extinct (*Turtle Taxonomy Working Group, 2017*; *Lovich et al., 2018*; *Rhodin et al., 2018*). Most turtle species are long-lived; our study taxa, the two species of WPT, live to 45 years (*Holland, 1994*), and female-biased adult mortality, with or without reduced recruitment, can have devastating consequences for long-lived species. Multiple studies have observed or proposed that male-biased populations result in population declines or extirpations of freshwater turtles (*Ceballos et al., 2016*; *Vanek & Glowacki, 2019*), and a long-term study of two wood turtle (*Glyptemys insculpta*) populations in Connecticut, USA documented sharp declines coinciding with human-mediated reductions in adult females (*Garber & Burger, 1995*). Reduced juvenile production, the ultimate result of increased female mortality, has also been found to result in freshwater turtle population declines or extirpations

(*Burgin & Ryan, 2008*; *Howey & Dinkelacker, 2013*). Such populations may persist for years, but they do so as part of the "living dead" (*Lovich et al., 2018*) that often characterize declining, long-lived species.

No published study has quantitatively examined these demographic issues for the WPT. We found that contemporary populations of both *E. pallida* and *E. marmorata* tend to be male biased. Because turtles, including the WPT, exhibit temperature-dependent sex determination, a conservation concern for many species is increasingly female biased sex ratios associated with climate warming. *Christie & Geist (2017)* demonstrated that *E. marmorata* from Lake County, CA had typical temperature-dependent sex determination for emydid turtles, with higher temperatures producing female-biased clutches, and that from 2009–2012 natural nests incubated in the wild produced 69% female hatchlings. Although only a single study, their work suggests that temperature increases from human-mediated climate change should be producing female-biased WPT primary sex ratios. If this is the case, the male bias that we document represents an even more severe increase in post-hatching female mortality than if the primary sex ratio were even.

Our results are less clear on changes in sex ratio over the last century. Our historical analysis of sex ratios is based on museum data, and these collections can be biased in many, sometimes subtle ways. Museum data may be biased if one sex or size class is more easily or commonly collected (*Gibbs & Steen, 2005*), as might occur in sexually or ecologically dimorphic taxa, or if collections come from different regions at different points in time (*Shaffer, Fisher & Davidson, 1998*). In our case, the museum data were drawn from across the range of both species (Fig. 1), which eliminates some bias that can occur with more limited sampling. Because female turtles spend more time on land than males and are therefore more vulnerable to incidental capture by museum scientists, any ecological bias should favor females over males, acting against the observed male bias in most collections. Fortunately, the sex ratios in contemporary population studies are based on aquatic trapping and hand-capture/snorkeling, both of which should be unbiased with respect to gender.

Assuming that museum and contemporary data represent relatively unbiased sampling efforts, both *E. pallida* and *E. marmorata* show a clear shift from essentially 1:1 in the early 20th century to male biased in contemporary samples. However, the sex ratio trends of *E. pallida* and *E. marmorata* differ in the last 20 years based on the current data sets (Fig. 4). While *E. marmorata* shows an increase in male bias over the last two decades, *E. pallida* shows a slight decrease in male bias. This may indicate that populations of *E. pallida* are returning to an even sex ratio, although no plausible mechanisms have been suggested for such a shift. An alternative explanation, first suggested by *Madden-Smith et al. (2005)* for parts of San Diego County, CA, is a form of population filtering that may be driving this trend. Particularly in southern California, where most recent population surveys have taken place, the remaining *E. pallida* sites are relatively remote with restricted human access, while the species has been extirpated from most areas with high anthropogenic impact. These last remaining populations may have experienced

less impact than was historically typical for the species, leading to a trend reflecting the survival and monitoring of an increasingly high proportion of viable populations. That is, the trends seen in *E. pallida* may represent the sequential elimination of human-impacted sites, rather than overall population recovery. Thus, while there are extant *E. marmorata* populations which have experienced anthropogenic impact, the *E. pallida* populations most impacted by humans have already been extirpated and those populations which remain are viable. Regardless of the cause of differences in trends over the last 20 years, the two species are now quite similar, and quite male-biased, in their sex ratios.

Our analyses provide strong, albeit correlational evidence that road proximity and density are associated with increasing male population bias. Our observation on the distance at which roads affect sex ratio matches closely with the distance females tend to travel from the water's edge, which ranges from a few meters up to a maximum of about 400 m. For both species, populations within 220 m of the closest road were male biased, while those further away were not (Fig. 6). Road density within 400 m of a study site, which models the likelihood that female mortality occurs during nesting road encounters, explained 27% of the variation in population sex ratios; the fitted linear regression of sex ratio as a function of road density predicts that the expected sex ratio of a population will change from 1:1 in roadless areas to 2:1 when ~7.5% of the area within 400 m is covered by road (Fig. 7). This is in contrast to previous studies which hypothesized that roads should not threaten small bodied pond turtles anywhere in the U.S. (*Gibbs & Shriver, 2002*), but is consistent with a recent threat assessment focused on California road mortality risk across reptiles and amphibians (*Brehme, Hathaway & Fisher, 2018*). Our analyses indicate that road density and proximity may lead to demographic changes of small bodied pond turtles like the WPT which could in turn drive population declines. Whether such changes are due to direct mortality from vehicles or the increased presence of human activity that roads inevitably incur is impossible to determine (*Garber & Burger, 1995*); both are likely contributing factors. In either case, a combination of under-road tunnels, barrier fences, and even simple "turtle crossing" signage coupled with community education and outreach could be effective strategies to stem the mortality of gravid females seeking nesting sites. Many of these road mitigation strategies have already been proven to be effective for turtles. A study in Presqu'ile Provincial Park, Ontario, Canada, found that both barrier fences and under-road tunnels reduced road mortality of turtles (*Boyle, 2019*). Additional studies support the efficacy of both of these strategies. Barrier fences effectively reduced road mortality of diamondback terrapins in Cape May County, New Jersey (*Ives-Dewey & Lewandowski, 2012*) as did under-road tunnels for painted turtles in Massachusetts (*Paulson, 2010*).

We also found a significant increase in carapace length over the last century in *E. marmorata* but not in *E. pallida*. Assuming that carapace length is a reasonable proxy for age, our results indicate that the average age of *E. marmorata* populations has increased over the last century while the average age of *E. pallida* has not. This is an unexpected difference, because we expected that the greater urbanization in the southern California range of *E. pallida* would result in decreased recruitment and overall population

senescence. In fact, the opposite appears to be the case. This could indicate that *E. pallida* has not had reduced recruitment in recent years, while *E. marmorata* has. Alternatively, the same "population filtering" discussed previously for southern California contemporary population studies may be driving this pattern. As a recent report indicates (*Madden-Smith et al., 2005*), non-native turtles and a large community of invasive predators have now essentially replaced native WPT in accessible San Diego County coastal habitats, and the few locations where WPTs still persist are free of many invasive predators. Consistent with this interpretation, most of the current *E. marmorata* population studies co-occur with bullfrogs (8/10 sites had bullfrogs present) while only half of the contemporary *E. pallida* populations had bullfrogs present (8/15 sites). However, the range of *E. pallida* actually has more iNaturalist sightings of bullfrogs per square km (density of bullfrog sightings in the range of *E. marmorata* = 0.002 bullfrogs/km$^2$, *E. pallida* = 0.008 bullfrogs/km$^2$, $p = 0.02$, two-tailed $t$-test) (Fig. S1). This is consistent with the interpretation that most of the impacted habitats in southern California with bullfrogs have already lost their pond turtles, and that the remaining *E. pallida* populations tend to be relatively remote sites without introduced predators. Although speculative, this line of reasoning suggests that invasive predator removal may be an important next step both in recovery of extant populations, and as a precursor to any WPT repatriation efforts.

## CONCLUSIONS

Natural history collections provide important historical baselines, including demographic trends, that are critical for declining species management. Based on our analyses, both shifts toward male sex-ratio bias and reduced recruitment have occurred in *E. marmorata*, consistent with a long, slow decline in population health. While male sex-ratio bias has also occurred in *E. pallida*, our data do not indicate that reduced recruitment is an ongoing threat for currently extant populations. Ultimately, the joint analysis of natural history collections as historical data with contemporary published and unpublished data sets shared with us through personal communications with WPT field researchers (B. Shaffer, P. Scott, R. Fisher, C. Brown, R. Dagit, L. Patterson, T. Engstrom, 2019, personal communications) allowed us to document demographic changes in both species which would have been otherwise impossible. Knowledge of these demographic changes provides important population trend information for current protections under consideration for both species. Our analyses indicate that roads and non-native predatory bullfrogs constitute significant threats to the long-term persistence of both species of WPT. Fortunately, both of these threats are, in theory, reversible. In combination with ongoing threat analyses and PVAs for both species (*Manzo et al., in press*), our results suggest that significant declines have occurred in both species, although the population health and stability of *E. pallida* is more fragile and in greater decline than *E. marmorata*.

Our assessment of demographic changes and their underlying potential mechanisms emphasizes how natural history collections can complement contemporary data to help identify shifts in population demographics and potential causative anthropogenic impacts responsible for these shifts.

## ACKNOWLEDGEMENTS

We thank all the field ecologists who provided data sets for our analyses including Tag Engstrom and Laura Patterson, and the USGS team that helped collect field data on turtles since the late 1990's. We also thank the curatorial staff who gave us access to natural history collections and provided their time and assistance, including Greg Pauly (Los Angeles Natural History Museum), Carol Spencer (Museum of Vertebrate Zoology), and Lauren Scheinberg (California Academy of Sciences). This is contribution number 744 of the U.S. Geological Survey Amphibian Research and Monitoring Initiative (ARMI). Any use of trade, firm, or product names is for descriptive purposes only and does not imply endorsement by the U.S. Government.

### Funding

Funding was provided by the UCLA Institute of Environment and Sustainability, the UCLA La Kretz Center for California Conservation Science, the Madelyn and Bruce Glickfeld Award, a mini-grant from the UCLA Center for the Advancement of Teaching, and a grant from the USFWS to UCLA. The San Diego Association of Governments (SanDAG) Transnet Program helped fund field work in San Diego County. Pre-listing funding was received by USGS from the USFWS and Ecosystems Mission Area of USGS. The funders had no role in study design, data collection and analysis, decision to publish, or preparation of the manuscript.

### Grant Disclosures

The following grant information was disclosed by the authors:
UCLA Institute of Environment and Sustainability.
UCLA La Kretz Center for California Conservation Science.
Madelyn and Bruce Glickfeld Award.
UCLA Center for the Advancement of Teaching.
UCLA.
San Diego Association of Governments (SanDAG).
USGS.

### Competing Interests

The authors declare that they have no competing interests.

### Author Contributions

- E. Griffin Nicholson conceived and designed the experiments, performed the experiments, analyzed the data, prepared figures and/or tables, authored or reviewed drafts of the paper, and approved the final draft.
- Stephanie Manzo conceived and designed the experiments, performed the experiments, analyzed the data, prepared figures and/or tables, authored or reviewed drafts of the paper, and approved the final draft.
- Zachary Devereux conceived and designed the experiments, performed the experiments, analyzed the data, prepared figures and/or tables, authored or reviewed drafts of the paper, and approved the final draft.
- Thomas Paul Morgan conceived and designed the experiments, performed the experiments, authored or reviewed drafts of the paper, and approved the final draft.
- Robert N. Fisher performed the experiments, prepared figures and/or tables, authored or reviewed drafts of the paper, and approved the final draft.
- Christopher Brown performed the experiments, analyzed the data, authored or reviewed drafts of the paper, and approved the final draft.
- Rosi Dagit performed the experiments, authored or reviewed drafts of the paper, and approved the final draft.
- Peter A. Scott conceived and designed the experiments, performed the experiments, prepared figures and/or tables, authored or reviewed drafts of the paper, and approved the final draft.
- H. Bradley Shaffer conceived and designed the experiments, performed the experiments, authored or reviewed drafts of the paper, and approved the final draft.

## Field Study Permissions

The following information was supplied relating to field study approvals (i.e., approving body and any reference numbers):

Permit for field tagging and tracking of western pond turtles by the Resource Conservation District of the Santa Monica Mountains is CDFW SC-604.

Permit for field tagging and sampling of western pond turtles by the USGS in Southern California is CDFW Scientific Collecting Permit (Entity) and MOU: SCP-838.

The Nature Conservancy gave permission to sample on the Dangermond Preserve which was also approved by the CDFW through the permit CDFW SC-2480.

## Data Availability

The raw museum data measurements and data collection are available in Dataset S1.

## Supplemental Information

Supplemental information for this article can be found online at http://dx.doi.org/10.7717/peerj.9248#supplemental-information.

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
