# Peer review of "Historical museum collections and contemporary population studies implicate roads and introduced predatory bullfrogs in the decline of western pond turtles"

_PeerJ, doi:10.7717/peerj.9248_

## Round 0.1 · original submission · Minor Revisions

· Academic Editor

Minor Revisions

Please pay close attention to the reviewer comments and modify your manuscript accordingly.

Reviewer 1 ·

Basic reporting

This was an extremely well-written paper and we enjoyed reading it (this review is from two of us reviewing together). The topic and questions are interesting and topical and the general approach to the data collection and analyses were sound.

We would have liked to see more thorough coverage of the literature on two points:
1) The history of American Bullfrog invasions. In particular, there needs to be some acknowledgement that this frog species is sympatric with a number of small-bodied turtles in eastern North America, and doesn’t seem to cause population declines there. Explaining why this is the case is beyond the scope of your paper, but a discussion of this conundrum would be helpful. Options for explaining the difference include 1) there is something different about the foraging behavior of invasive bullfrogs (there is literature to support this; invasive bullfrogs are more bold, for example), and/or 2) there is something different about the behavior of turtles in historically bullfrog-free areas that makes them more susceptible to bullfrog predation. However you approach this, we suggest adding text to clarify that it’s not as simple as “bullfrogs are bad for small-bodied turtles”.
2) Can you more thoroughly consider the literature on road effects on turtle sex ratios? The evidence is less clear than the Introduction makes it sound. We appreciated the statements that the effects of roads on sex ratios are still unclear, but this is still the working assumption of your study. We would have liked to see inclusion of some more papers that contradict the “roads cause male-biased populations” narrative. Some options (there are others if you prefer) include:

Rytrwinski and Fahrig (2014, PlosONE) found no male-bias associated with painted turtle populations’ proximity to roads: https://www.ncbi.nlm.nih.gov/pmc/articles/PMC4032323/

Carstairs et al. (2019; Canadian Field Naturalist) found that road mortality of turtles in Ontario, Canada was not sex-biased, but that there is seasonal variation in mortality of males and females (https://s3.amazonaws.com/academia.edu.documents/58867726/CFN-12_1908_Carstairs_4.pdf?response-content-disposition=inline%3B%20filename%3DRevisiting_the_hypothesis_of_sex-biased.pdf&X-Amz-Algorithm=AWS4-HMAC-SHA256&X-Amz-Credential=AKIAIWOWYYGZ2Y53UL3A%2F20200221%2Fus-east-1%2Fs3%2Faws4_request&X-Amz-Date=20200221T151315Z&X-Amz-Expires=3600&X-Amz-SignedHeaders=host&X-Amz-Signature=96b0f8df01416eca4788ea4c113d14edaafd061082be5f5c3da4b274f40b1877 )

This concern about over-stating the strength of the evidence for road effect on turtle sex ratios also applies to the discussion. Your data are compelling, but they are (as you say) an association. Be very careful about linking sex ratios to roads; the evidence is not clear.

Lines 405-408 - there is now a large body of literature evaluating the effectiveness of road mitigation measures. Can you cite some of these studies to acknowledge that your statement about what could work well is not a new one?

Experimental design

The experimental design is sound.

Validity of the findings

The findings appear valid based on the presented data.

Please add test statistics to the methods. Currently there are several places where a p-value is reported in the text but the test statistics are not. We found these elsewhere (i.e. in the figure captions) but it would be easier for the reader if they were all presented clearly, together.

Figures 2 and 3: these maps need an inset showing where the study area is in relation to North America (not all your readers will be from the US... we aren't!). Then zoom the larger figure area in to the area of interest so that the locations can be seen more clearly. Please add some labels (these could include "United States of America", "Mexico", "Baja California", "California", "Pacific Ocean", etc.).

We suggest you combine Figures 4 and 5 into a single figure and combine 6 and 7 into a single figure, and add labels to indicate the two species. Also, is there anything you can do to make these figures more visually effective? There is nothing technically wrong with them, but the Large Black Blobs indicating the contemporary data look quite odd.

Also on Figure 4 and 5, consider adding labels for Male and Female at the ends of the y-axis, because you are plotting sex as a binary variable in this figure. The graded axis is then showing changes in sex ratio, so change the axis label to Sex Ratio, but the points indicating Sex are binary).

Is there a reason that the logistic regressions in Figures 6 and 7 have 95% CIs but Figures 4 and 5 do not?

Figure 9 - could be more clear if you use the same phrasing you used in the figure caption (which we found very clear!) for the x-axis label

Additional comments

This is a well-written and interesting paper and all the comments we've made above and below are around tidying and clarifying it; they should be relatively easy to do.

You use WPT as a catch-all for two species. WPT is a common name for a species that has now been split in two, and these now need new, distinct common names. If those don't exist yet, you still need to find a way to clarify that you are discussing two different species, pooled into one group (for all the excellent reasons you provided in Lines 294-295 - this approach makes sense, it just needs more clear language).

Minor suggestions to improve clarity:
- Add info about study location (south-western United States) in the abstract.
- Line 107 - can you give some information about the regions in which this evidence was collected? There is not clear evidence of global shifts in turtle sex ratios, and the evidence is equivocal among studies.
- Lines 211-212 "somewhat independent analyses" - We weren't quite sure what you meant by this so we don't have a good suggestion to replace it. Can you revise to clarify?
- Consider maintaining a clear order in your consideration of the two species? In Line 347, you discuss E. marmorata first; in Line 370 you start a comparison with E. pallida. It will be easiest for the reader to keep up if you pick an order, and use it consistently in the text and also in the figures.
- You use the word "healthy" to refer to populations in several places. "Healthy" is 1) difficult to quantify, and 2) arguably isn't determined by road mortality or predation rates. Do you mean "viable"?
- Lines 304-310 - move to methods.
- can you provide a bit more detail about the roads in your study? are they all the same? high traffic? Low traffic? Same speed limits?
- Line 332: "endangered" has a specific meaning under IUCN and US guidelines for species assessment. Do you mean "threatened" (encompasses all IUCN risk categories)
- can you standardize your p-value reporting? Sometimes you use "p = ", sometimes "p-value..." etc.


We hope these suggestions are helpful, and we wish you the best of luck with this study.

Reviewer 2 ·

Basic reporting

No comment.

Experimental design

No comment.

Validity of the findings

No comment.

Additional comments

I found this to be rigorous and well-designed, -executed, and -written study. I only have a few minor editorial comments.

1) Line 44 - I suggest inserting the scientific name of both western pond turtle species "(Emys marmorata and E. pallida)" in parentheses after "...two paripatrically distributed species". Currently, the generic abbreviation (E.) is used on line 55 with no prior reference to the full generic name.

2) Line 97 – “Fig. 2” is referenced here without a prior reference to “Fig. 1” (first referenced on line 133). Shouldn’t the first mention of figures in the text be in numerical order?

3) Line 270 – Change “shows” to “show” (“data” is a plural noun and should be followed by a plural verb).

4) Line 329 – Change “lower” to “higher” or “greater” (as the text above and supporting graphs indicate).

5) Line 397 – Replace “changes” with “change”.

Reviewer 3 ·

Basic reporting

The literature cited section has some issues. There are several citations missing from the text and several from the Literature Cited Section itself. Several citations are also miss labeled within the text of the document.

Lines 88 and 91 - Fritz et al. 2012 is listed as Fritz et al. 2011 in the Literature Cited Section

Line 376 - Madeen-Smith (2005) should be Madden-Smith et al. (2005).

Line 112 - Spellerberg, 1998 is missing from the Literature Cited Section

Line 216 - Belli, 2015 is missing from the Literature Cited Section.

The following could not be found within the text of the document:

Bureau of the Census 2012

Manzo et al. 2019

Nyholf, 2013

Rathburn et al. 1992

Steen et al. 2006

Rhodin et al. 2018 is not cited correctly within the Literature Cited Section. Short hand was used and I do not believe that to be an accurate representation of the cited work.

Everything else seems to follow the journals format and is in good form.

Experimental design

I found the experimental design to be quite interesting and carried out well. Truly, the authors did a fine job in stating their premise, carrying out the experimental design, and explaining their results in a clear and concise manner. In fact, the issues that I was going to bring up where actually brought up by the authors themselves within the text as they defended the reasoning around what they did and why they did it.

Validity of the findings

Valid and publishable. This paper is a timely and important one for the two WPT species. Invasive species can be extremely detrimental to native species populations. This has been proven time and time again. It is nice to see some work going into proving that the invasive bullfrog is having a measurable effect on the WPT populations. Now maybe something can be done about the frogs. Population monitoring and culling of the invasive predator should be started at these known WPT sites and corresponding WPT populations monitoring should continue to show what the affect of reduced predation is over the subsequent years.

Additional comments

Nice paper. I enjoyed reading it. Pay more attention to the little things like the Literature Cited Section as it is always something that authors seem to relax their guard on and is typically one of the first things more editors I know look at.

---

## Round 0.2 · accepted · Accept

· Academic Editor

Accept

Thank you for your efforts in responding to reviewer comments and modifications to your manuscript. I appreciate it, and the document has been improved.